# Scaling up of tsetse control to eliminate Gambian sleeping sickness in northern Uganda

**Andrew Hope**[1]\*, **Albert Mugenyi**[2]\*, **Johan Esterhuizen**[1], **Inaki Tirados**[1], **Lucas Cunningham**[1], **Gala Garrod**[1], **Mike J. Lehane**[1], **Joshua Longbottom**[1], **TN Clement Mangwiro**[3], **Mercy Opiyo**[1,4], **Michelle Stanton**[1], **Steve J. Torr**[1]\*, **Glyn A. Vale**[5,6], **Charles Waiswa**[2], **Richard Selby**[1]

**1** Liverpool School of Tropical Medicine, Liverpool, Merseyside, United Kingdom, **2** Coordinating Office for Control of Trypanosomiasis in Uganda, Kampala, Uganda, **3** Bindura University of Science Education, Bindura, Zimbabwe, **4** Barcelona Institute for Global Health, Hospital Clinic, Barcelona, Spain, **5** Southern African Centre for Epidemiological Modelling and Analysis, University of Stellenbosch, Stellenbosch, South Africa, **6** Natural Resources Institute, University of Greenwich, Chatham, United Kingdom

\* andrew.hope@lstmed.ac.uk (AH); albertmug@yahoo.com (AM); steve.torr@lstmed.ac.uk (SJT)

**Data Availability Statement:** All relevant data are within the manuscript and its Supporting Information files.

## Abstract

### Background

Tsetse flies (*Glossina*) transmit *Trypanosoma brucei gambiense* which causes Gambian human African trypanosomiasis (gHAT) in Central and West Africa. Several countries use Tiny Targets, comprising insecticide-treated panels of material which attract and kill tsetse, as part of their national programmes to eliminate gHAT. We studied how the scale and arrangement of target deployment affected the efficacy of control.

### Methodology and principal findings

Between 2012 and 2016, Tiny Targets were deployed biannually along the larger rivers of Arua, Maracha, Koboko and Yumbe districts in North West Uganda with the aim of reducing the abundance of tsetse to interrupt transmission. The extent of these deployments increased from ~250 km² in 2012 to ~1600 km² in 2015. The impact of Tiny Targets on tsetse populations was assessed by analysing catches of tsetse from a network of monitoring traps; sub-samples of captured tsetse were dissected to estimate their age and infection status. In addition, the condition of 780 targets (~195/district) was assessed for up to six months after deployment. In each district, mean daily catches of tsetse (*G. fuscipes fuscipes*) from monitoring traps declined significantly by >80% following the deployment of targets. The reduction was apparent for several kilometres on adjacent lengths of the same river but not in other rivers a kilometre or so away. Expansion of the operational area did not always produce higher levels of suppression or detectable change in the age structure or infection rates of the population, perhaps due to the failure to treat the smaller streams and/ or invasion from adjacent untreated areas. The median effective life of a Tiny Target was 61 (41.8–80.2, 95% CI) days.

**Funding:** Funding for this research was provided by the Bill and Melinda Gates Foundation (www. gatesfoundation.org), with grants awarded to MJL (Grant ID#:1017770) and SJT (Grant ID#: OPP1104516), and the UK Biotechnology and Biological Sciences Research Council with grants awarded to SJT (BB/S01375X/1, BB/P005888/1, BB/L019035/1) The funders had no role in study design, data collection and analysis, decision to publish, or preparation of the manuscript.

**Competing interests:** The authors have declared that no competing interests exist.

## Conclusions

Scaling-up of tsetse control reduced the population of tsetse by >80% across the intervention area. Even better control might be achievable by tackling invasion of flies from infested areas within and outside the current intervention area. This might involve deploying more targets, especially along smaller rivers, and extending the effective life of Tiny Targets.

## Author summary

Gambian human African trypanosomiasis (gHAT) is a neglected tropical disease caused by *Trypanosoma brucei gambiense* transmitted by tsetse flies (*Glossina*). Uganda's strategy to eliminate gHAT includes the deployment of Tiny Targets, comprising insecticide-treated panels of cloth which attract and kill tsetse. Our data from a network of monitoring traps assessed how increasing the intervention area from ~250 km$^2$ to ~1600 km$^2$ affected the degree of control. Inspection of deployed targets indicated their effective lifespan. Targets reduced tsetse abundance by >80% beside the rivers where they were deployed but had no clear effect on adjacent rivers where targets were absent. As the intervention area increased, so did the extent of the area controlled. We did not deploy targets along the smaller rivers so that, as expected, the tsetse population was not eliminated. Our findings suggest that the population was sustained at low levels by invasion of tsetse from untreated parts of the drainage system. The average effective life of targets was ~60 days as against the ~180 days for targets deployed in Kenya. This discrepancy is attributable, in part, to the Uganda targets being removed by seasonal floods. While the level of control achieved is already more than sufficient to interrupt transmission of gHAT, even better control would be achieved by increasing the coverage of the drainage system.

## Introduction

Sleeping sickness (human African trypanosomiasis, HAT) is caused by sub-species of *Trypanosoma brucei* spread by tsetse flies (*Glossina*). In the 20[th] Century there were three major epidemics of HAT, the most recent being in the late 1990s when up to >35000 cases were reported annually [1]. WHO declared the aim of eliminating HAT as a public health problem by 2020 and the complete interruption of transmission by 2030. Over the past 20 years, the number of cases reported globally has declined by >95%, with <2000 cases reported annually since 2017. WHO has formally validated the successful elimination of HAT as a public health problem in Côte d'Ivoire and Togo, where cases of HAT have been <1 case per 10000 inhabitants in all districts during a five-year period.

Most (>95%) cases of HAT are caused by *T. b. gambiense*, giving rise to Gambian HAT (gHAT), transmitted by riverine species of tsetse. Sub-species of *Glossina fuscipes* are the major vectors across a large swathe of Central Africa, extending across South Sudan, Uganda, the Democratic Republic of Congo (DRC), Central African Republic (CAR) and Chad [2]. Together, these countries contributed >90% (27296/32275) of all cases of gHAT reported globally in the period 2011–2020 [1].

The current low incidence of the disease has been achieved largely through the mass screening and treatment of the human population. However, modelling and empirical evidence suggests that vector control can make an important contribution to achieving the elimination

goals [3,4]. The development of effective and affordable methods of tsetse control makes such an integrated approach feasible. In places where cattle are present in adequate densities (>10 cattle/km$^2$), treatment of cattle with pyrethroids to control tsetse can be highly cost-effective [5,6]. In areas with lower densities of cattle, the deployment of insecticide-treated targets which attract and kill tsetse is recommended [7,8].

An important consideration in the use of insecticide-treated cattle and targets is their spatial distribution. The riverine vectors of gHAT can displace at ~300 m/day [9] modulated by local topography [10]. This degree of mobility means that small scale operations, covering say a single small village within 4 km$^2$, will be ineffective because tsetse from neighbouring uncontrolled areas invade rapidly, so ensuring no perceptible change in the numbers of tsetse [11]. The corollary is the need to treat larger areas, of the order of ~50 km$^2$ or more. However, this problem is offset by several considerations. First, since tsetse breed very slowly [12], the population of the large areas can be controlled by killing only 3.5% percent of the adult females per day, so that the baits need to be deployed only sparsely within the habitat. In that regard, the high mobility of tsetse becomes an advantage, since the flies can locate sparsely-placed baits. Second, the riverine and lake-shore habitats of the riverine vectors of gHAT form only a small proportion of the land surface, so reducing further the required number of targets. Third, to interrupt the transmission cycle of gHAT it is not necessary to eliminate tsetse completely–epidemiological modelling suggests that reducing the population density by ~70% can be sufficient [13]. Finally, the so-called Tiny Targets, measuring only ~0.1 m$^2$ [14], that are recommended for riverine tsetse, are more cost-effective than earlier designs [15]. Such targets have been particularly effective against riverine species of tsetse in the gHAT foci of Uganda [13], Chad [4], Guinea [16], DRC [17] and Côte d'Ivoire [18]. Epidemiological analyses have also shown that deployment of Tiny Targets reduced the incidence of gHAT in Guinea [19], Chad [4] and Uganda [20].

Any method of controlling trypanosomes and tsetse has its own strengths and weaknesses. For vector control the biggest problem is arguably the invasion of tsetse from uncontrolled areas nearby. Thus, even with the economies associated with the control of riverine tsetse by Tiny Targets, it is usually too costly and impractical to avoid this problem by eliminating tsetse within the invasion sources because such sources are commonly very extensive in comparison to the foci where gHAT occurs. A better understanding of how the level of suppression is affected by the degree to which control operations cover the drainage systems would provide a rational basis for optimising strategies for controlling tsetse. The general principles governing the relationships between the scale of control operations, density of targets and mobility of tsetse are understood but we have little empirical evidence to guide the design of control campaigns applicable to specific ecological and epidemiological settings.

Theoretical modelling of the impact of tsetse control on the transmission of gHAT has suggested that reducing populations by >70% will reduce R0 below 1 [13]. Similarly, modelling of gHAT in DRC predicted that >60% reductions will lead to the interruption of transmission [21]. Empirical studies of the impact of Tiny Targets on transmission suggest that reductions of 80% in Guinea reduced transmission [16], and a <100% reduction in Chad interrupted it [4]. These and similar theoretical and empirical studies all suggest that transmission of gHAT can be interrupted, or at least reduced, without eliminating the population of tsetse. However, it is not clear whether there is a unique and universal reduction threshold above which elimination of transmission can be achieved. In practice, the reported reductions in tsetse density for the above interventions are pertinent to their localities, and it seems likely that there will be spatial variation in the reductions achieved. Accordingly, one of the motivations for the present work was to quantify spatial variation in the apparent reductions in tsetse abundance caused by deploying Tiny Targets, focussing on two research questions.

First, what is the impact of tsetse control within the drainage systems that form the main habitat? Previous studies of tsetse in Uganda suggest that tsetse are less abundant nearer the sources of rivers where the small streams are seasonal and riverine vegetation is sparse [22]. We might therefore expect that vector control will have a greater impact in upstream areas of drainage system. Moreover. riverine tsetse move along drainage systems [23] and hence we might also expect that the problem of invasion might be greater in downstream sections of the system. Second, does vector control on one river affect tsetse populations on an adjacent untreated river? If tsetse move largely along rivers and not across the interfluve that lies between them, then we would not expect vector control to affect populations on adjacent untreated rivers.

Tsetse control increases the mortality rate of a tsetse population which reduces the abundance of flies and their mean life expectancy [24]. For instance, use of insecticide-treated targets to control savanna tsetse in Zimbabwe reduced the abundance and the mean age of the tsetse population [25]. Given that the maturation of trypanosomes in tsetse takes 10–20 days [26] [27], reductions in life expectancy may also lead to a reduction in infection rates. Another aim of our study was therefore to assess whether large-scale interventions against tsetse had an impact on the age structure and trypanosome infection rates of tsetse populations.

Uganda was one of the first countries to use Tiny Targets successfully in an area affected by gHAT [13]. Between 2011 and 2012 (Phase 1), targets were deployed across five separate $7 \times 7$ km blocks of Arua and Maracha districts in north west Uganda. In early 2013 (Phase 2), the control operation expanded to produce a single operational area of 500 $km^2$. The operation resulted in a >90% reduction in numbers of tsetse caught by traps in the areas where targets were deployed. From January 2015 onwards (Phase 3), tsetse control in north west Uganda was expanded further to form an operational area of ~2500 $km^2$ covering most of the HAT-affected areas of Arua, Maracha, Koboko Yumbe and Moyo districts [28]. The efficacy of these control operations has been monitored consistently using a network of tsetse traps.

In previous papers, we have reported the impact of these control operations on tsetse populations [13], their economic cost [15], the perceptions that local people have about baits [29], and the impact on gHAT cases [20]. Here we use data from entomological monitoring to assess how the various scales (50–1600 $km^2$) of control operations conducted across a contiguous area covering parts of Arua, Maracha, Koboko and Yumbe districts, impacted on the distribution and abundance of tsetse populations within and outside the intervention areas. In particular, we address the following four questions:

1. What was the impact of small-scale (50 $km^2$) interventions on the abundance of tsetse in neighbouring untreated areas?

2. What was the impact in the untreated rivers adjacent?

3. Did expansion of tsetse control operations produce higher levels of suppression within the control area and alter the population age structure and infection rates of tsetse?

4. What is the operational lifetime of Tiny Targets when deployed in a large-scale tsetse control programme?

## Materials and methods

### Ethics statement

The research did not involve any human participants, domestic livestock or wild vertebrate animals. The data were produced under the auspices of the Uganda National Council of Science and Technology ethics board ("Targeting tsetse: use of targets to eliminate African

sleeping sickness" Ref. Number HS 939) and a Memorandum of Understanding between the Uganda Trypanosomiasis Control Council (UTCC) and the Liverpool School of Tropical Medicine ("Tsetse control programme using insecticide-treated Tiny Targets in the West Nile HAT foci of Uganda (Moyo, Koboko, Arua, Maracha and Yumbe", ADM.7/166/01).

## Research questions

1. *What was the impact of small-scale (50 km² ) interventions on the abundance of tsetse in neighbouring untreated areas*? We hypothesized that tsetse are confined to rivers and that the flies move mainly along rather than between rivers. Accordingly, we expected that small-scale interventions applied to a particular river system would impact on tsetse populations on that river but not adjacent ones. To test this hypothesis, we compared the catches from traps deployed in a transect across the original small (7 × 7 km) intervention blocks to assess whether the overall impact varied according to local hydrography.

2. *What was the impact in untreated rivers adjacent*? We hypothesized that interventions on one river would not have an impact on adjacent ones. To test this hypothesis, we monitored the abundance of tsetse in rivers adjacent to those where Tiny Targets were deployed, to assess whether the impact of targets was extended beyond the deployment rivers.

3. *Did expansion of tsetse control operations produce higher levels of suppression within the control area and alter the population age structure and infection rates of tsetse*? As the extent of control operations expanded, we expected that improved levels of control might be achieved because the proportion of the area subject to re-invasion would be reduced. To elucidate this, we compared the catches from a network of traps deployed across the intervention area. We hypothesized that traps near the edge of the operational area would have catches greater than those at the centre if the operational area were invaded at a significant rate from outside, as against the redistribution of flies between the treated and untreated rivers within the operational area. In addition to reducing the catch of tsetse, we also assessed whether the age structure of the tsetse population was altered; since targets increase the local death rate of tsetse we predicted that the control measures would reduce not only the numbers of tsetse but also the mean age of the population if invasion were not important. This prediction is in accord with the theoretical [30] and practical [25] aspects of target performance.

4. *What is the operational lifetime of Tiny Targets when deployed in a large-scale tsetse control programme*? As the intervention area increased, the operation transitioned from being an externally managed research project to being managed by the Co-ordinating Office for Control of Trypanosomiasis in Uganda (COCTU) and implemented by District Entomologists rather than researchers and technicians. Monitoring the deployment of targets and their impact on tsetse populations allowed us to assess whether scaling up affected the overall quality of target deployment and the impact on tsetse populations.

## Study area

All field studies were carried out in Arua, Maracha, Koboko and Yumbe districts in North West Uganda where gHAT has been a long-standing health problem—with 2928 cases being reported in the period 2001–2010 [31]. These districts are within the West Nile region, bordering South Sudan and the Democratic Republic of the Congo, and mostly comprise small-scale arable farms producing cassava, millet, peanuts and tobacco, with relatively few cattle and pigs [32]. There are two perennial rivers (Enyau, Kochi) and numerous smaller (<3 m wide) tributaries and streams, some of which cease flowing during the dry season (January-March). Narrow (~2–5 m) bands of natural vegetation (e.g., *Cynometra alexandri*, *Entada abyssinica*, *Acacia seyal*) along the banks of the rivers and streams provide habitat for tsetse. Wild

mammalian hosts are rare in the area but Nile monitor lizards are present and form part of the diet of tsetse, along with humans, cattle and pigs [33].

## Deployment of Tiny Targets

Tiny Targets [14,34] have been used in North West Uganda since 2011 [13] with the scale of control being increased in three phases (Fig 1). For each phase, Tiny Targets were deployed along the rivers and larger streams at 100 m intervals on both banks giving a density of 20 targets/km along treated rivers. In general, targets were deployed at six-month intervals because previous work had shown that the performance of the insecticide declined markedly after six months [13]. Between November 2011 and November 2012, Phase 1, targets were deployed in five separate blocks, each block being a $7 \times 7$ km square, producing a nominal total coverage of ~250 km$^2$ (Fig 1A). From December 2012, Phase 2, targets were deployed over a larger area of, nominally, 500 km$^2$ which consisted of a convex hull encompassing the initial five blocks (Fig 1B). In 2014, the first deployment was carried out successfully in January-March but the second deployment, conducted in July 2014, was incomplete with targets being deployed in the original (Phase 1) Inve, Aiivu and Kubala blocks but not elsewhere (see S1 Text). Finally, between January 2015 and December 2016, Phase 3, the extent of operations expanded to a nominal area of 1636 km$^2$ (Fig 1C). In Phases 1, 2 and 3, the total number of Tiny Targets deployed every six months were ~1300, ~2900 and ~14100 targets, producing overall densities of 5, 6 and 9 targets/km$^2$, respectively (Fig 1D-1F).

## Entomological monitoring

Tsetse populations were monitored using pyramidal traps [35]. The numbers of monitoring sites expanded with the extent of the control operation (Fig 1). For Phase 1, pairs of traps, 100 m apart from each other, were deployed at the centre, edge and ~2 km outside the $7 \times 7$ km blocks (S1A Fig), so providing a total of 50 monitoring traps in or close to areas where targets were deployed. For Phase 2, a further 14 traps were added to the existing ones to provide 64 traps across the 500 km$^2$ intervention area. For Phases 1 and 2, 14 traps were also deployed along the Kochi river in Koboko district (northern most traps in Fig 1A & 1B) where no targets were deployed. These traps provided a measure of the underlying seasonal variation in the abundance of tsetse in the absence of Tiny Targets. Finally, in Phase 3, a network of 82 traps was used to monitor tsetse across 1636 km$^2$. For Phase 3, the expansion of tsetse control operations included the Kochi river but traps were routinely operated along sections of the Oluffe river in Maracha district, and the Enyau river in Arua district where targets were not deployed. These sites were in areas that had been originally subject to control during Phase 2 and the catches provided a measure of the response of a tsetse population to the cessation of control.

For all phases, daily catches from traps were collected, counted and identified to species and sex, for 5–20 days per month. Tsetse collected from a sub-sample of 15 traps (Fig 1C, stars) were regularly dissected to provide estimates of infection status and age in areas with or without Tiny Targets.

## Age and infection status of captured tsetse

Captured tsetse were transported from the trap sites to a laboratory in Arua in humid cool boxes to reduce their mortality. At the laboratory, live female tsetse were dissected to assess their ovarian category on a scale of 0–7 [36] and infection status [37]. For the latter, we examined the midgut, mouthparts, and salivary glands for the presence of trypanosomes at x400 using a compound-microscope with a dark-field filter. Classically, the identification of trypanosomes in tsetse is based upon their location: trypanosomes in the salivary glands and in

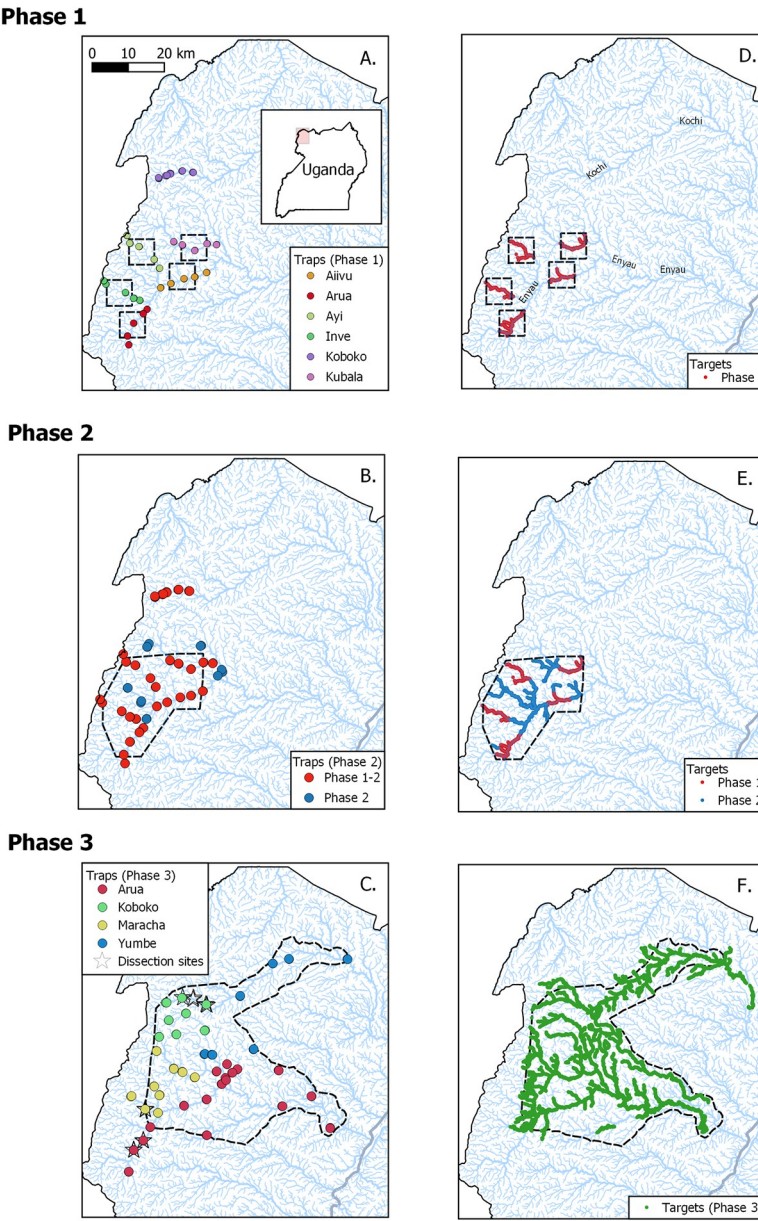

**Fig 1. Extent of planned tsetse control operations (black polygons with dashed lines) and locations of monitoring traps and Tiny Targets deployed between 2011 and 2016.** (A) Phase 1 (November 2011- November 2012): five square blocks, each 7 × 7 km. (B) Phase 2 (December 2012 –December 2014): single polygon of 500 km². (C) Phase 3 (January 2015 –December 2016): single polygon of 1636 km². For C only, monitoring traps with stars indicate sites where tsetse were collected for dissection. D-F show, in (D) red, (E) red & blue or (F) green where targets were deployed in Phases 1–3; the untreated rivers and streams are shown in pale blue. Base layer of rivers is derived from [22].

the midgut are presumed to be *T. brucei*, midgut and mouthpart infections are T. *congolense* and mouthparts only are *T. vivax* [37]. However, without additional PCR-based analyses [38] this method cannot distinguish co-infections or immature infections with *T. brucei* and *T. congolense* where the infection is confined to the midgut only. We did not perform PCR-based analyses of the dissected tsetse and so have not used location of infection to infer the species of trypanosome.

## Monitoring Tiny Targets

The locations of all Tiny Targets and monitoring traps were recorded with a global positioning system (GPS, Garmin eTrex) as the targets and traps were deployed. These data were mapped using QGIS, an open-source geographical information system (GIS). The GIS also included a shapefile of the local river network derived from 30 m resolution digital elevation model (DEM) data obtained from the Shuttle Radar Topography Mission (SRTM) and categorised using the Strahler stream order [22]. The stream order was used to define large (order>6), medium (4–6) and small (<4) rivers to aid visualisation of the main rivers and their tributaries.

In 2015, we randomly selected ten 1-km sections of rivers in each district where targets had been deployed; sections were excluded if they were (i) >1 km (~10 min walk) from a vehicle access point, (ii) included a village or (iii) required walking for >30 mins to cover the length of the river. Each section of river had ~20 targets. GPS records of each target's location and its date of deployment were used to estimate how long each target had been in the field. The condition of each target located was recorded. Targets that were present, upright and undamaged were regarded as functional.

## Experimental design

**Research question 1.** To assess the impact of interventions on neighbouring treated areas, we compared temporal changes in catches from traps deployed along a transect extending upstream-downstream of the intervention (S1A Fig). We predicted that catches would decline following the deployment, and that the decline would be greater at the centre of the intervention rather than at the edges. We also predicted that impact on the edges or outside the intervention area would be greater in upstream sections of the drainage system. To assess whether any temporal changes in catch were related to seasonal or other environmental change, we compared catches in the intervention area with those from the Kochi river where no targets were deployed (Fig 1). There we predicted that there would be no significant decline in catches when Tiny Targets were deployed elsewhere.

**Research question 2.** To assess whether deployment of targets on one river had an impact in the untreated rivers adjacent, we compared temporal changes in catches as targets were deployed on the Ayi, Inve, Oluffe and Kochi rivers. We predicted that catches would decline on the Ayi and Inve rivers following the deployment of targets in Phase 1. The Oluffe river lies between the Ayi and Inve and targets were deployed there in Phase 2. We predicted that catches would decline there during Phase 2 but not Phase 1. To assess whether any observed changes on these rivers were due to seasonal effects, we compared catches in intervention areas with those from traps on the Kochi river where no targets were deployed. A limitation to this design is that monitoring on the Oluffe river started later than that on the other rivers and hence we do not have data to compare catches on all four rivers before any targets were deployed.

**Research question 3.** We used an expansion of tsetse control operations across four districts (Phase 3) to assess whether large-scale tsetse control interventions produce higher levels of suppression and alters the age structure and infection rates of the tsetse population. The expansion of operations included our original non-intervention monitoring sites on the Kochi river. Accordingly, we selected sites on the Enyau and Oluffe rivers where targets were not deployed during Phase 3. We compared temporal changes in catch from monitoring traps in each district with those from the traps in non-intervention areas on the Enyau and Oluffe. We analysed samples of tsetse from monitoring traps in areas with and without targets to compare age structure and infection rates. We predicted that in areas where targets were deployed

catches, mean age and infection rates would decline. A limitation to this design is that we do not have data to compare simultaneously the abundance, age and infection status across all sites before any targets were deployed.

**Research question 4.** During the expansion of tsetse control operations (Phase 3), we quantified the rate that targets were lost or damaged following their deployment. We predicted that the numbers lost and/or damaged would increase with time since deployment. A limitation to the design is that logistical constraints restricted our ability to select targets for monitoring, as detailed above.

## Statistical analyses

All analyses were carried out using the open-source statistical software R [39].

## Tsetse abundance

To assess the impact of Tiny Targets on the abundance of tsetse, we compared the catches from monitoring traps before and after the deployment of targets. We predicted a significant decline in catch from traps in the vicinity of targets but no change for traps deployed on untreated rivers (e.g., Kochi river in Phases 1–2, sections of the Enyau and Oluffe rivers in Phase 3). We used the 'glmmADMB' package to fit generalized linear mixed models (glmm) to catch data using a negative binomial data distribution and a log link function. To visualise the overall trends following the deployment of targets, the site and day of capture were specified as random effects and months as a fixed effect to estimate mean daily catches and their 95% Confidence Intervals for each month. To assess the statistical significance of changes in catch, we compared catches before and after targets were deployed using the 'glht' function from the 'multcomp' package. Mean daily catches for months or years after deployment are expressed as a proportion of the catch before targets were deployed. That proportion is termed the Catch Index; an index significantly less than unity indicates a decline in catch.

## Ovarian categories and infection rates

We compared the age structures of tsetse caught from areas with and without targets using Chi-squared tests. To assess differences in infection rates, we fitted data to a general linear model (glm) with a binomial error distribution and a logit link function. The statistical significance of explanatory factors and variables was assessed by Analysis of Deviance (ANODEV) using the 'anova' function.

## Target condition

We also fitted data to a glm with a binomial error distribution to assess variation in the proportion of targets that were functioning correctly. Duration after deployment was specified as an explanatory variable and the statistical significance of changes in the proportion of functioning targets with time was assessed by ANODEV. The effective half-life of targets was estimated using the dose.p function from the 'MASS' package.

# Results

## Impact of small-scale interventions on neighbouring untreated areas

Following the deployment of Tiny Targets in relatively small (7 × 7 km) areas, the abundance of tsetse declined within and outside those areas (Fig 2). For all locations on rivers where targets were deployed, the catches after the target deployment (2012–2014) were significantly lower (P<0.001 for all comparisons) than those before (2011) whereas there was no significant

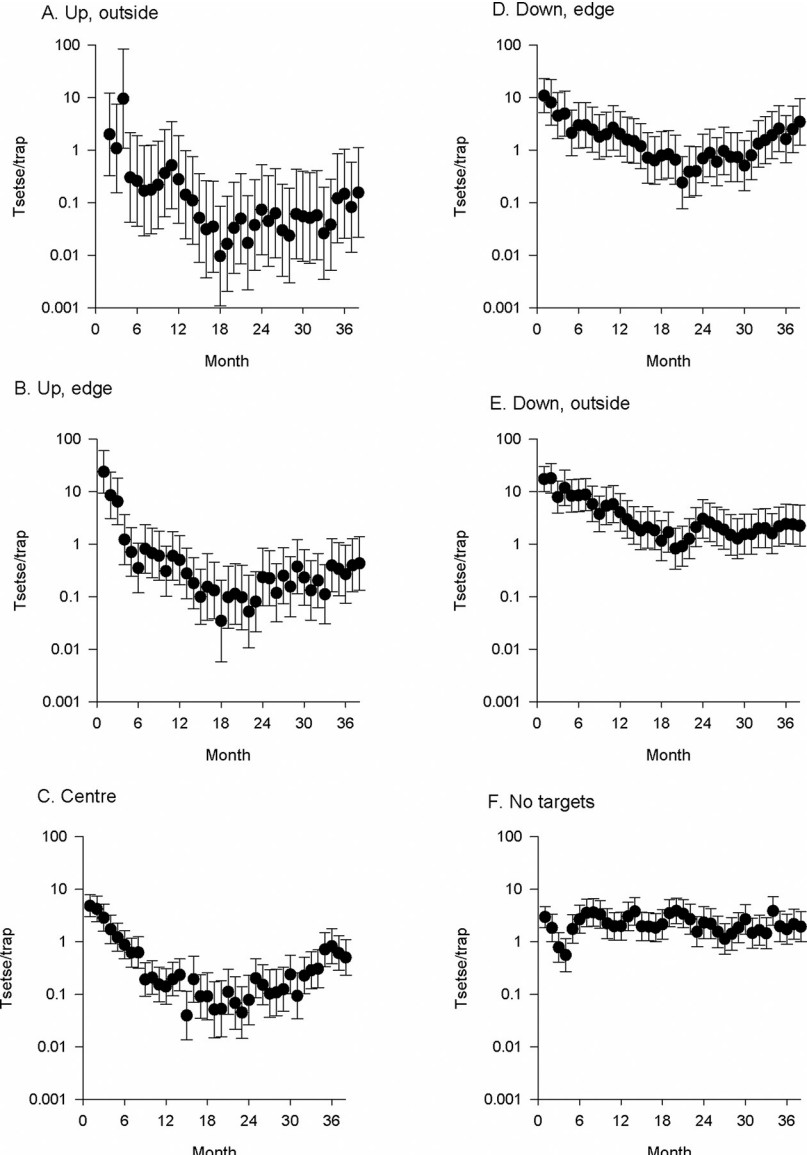

**Fig 2. Mean daily catch of tsetse from traps transecting Phase 1 intervention blocks.** For each intervention block, pairs of traps were located at sites (A) upstream and outside, (B) on the upstream edge, (C) at the centre, (D) downstream edge and (E) downstream and (F) outside.

difference where targets were absent (P = 0.12–0.99) (S1A Table). The decline in catch varied according to the location of the traps relative to the intervention area. For instance, traps on upstream and outside the block produced the lowest catches, with mean daily catches being 0.03–0.18 tsetse/trap, whereas those downstream and outside produced daily catches of 2.03–4.68 tsetse/trap (S1A Table).

To show more clearly the trend along the transect from outside-upstream to outside-downstream, we expressed each monthly catch (Fig 2) for the period January 2012-December 2014 as a proportion of the mean catch for the period September-October 2011 when no targets were deployed. The results (Fig 3) show that the median catch indices were 0.03–0.04 for traps upstream or at the centre of the block compared to 0.12–0.13 for the downstream traps. In

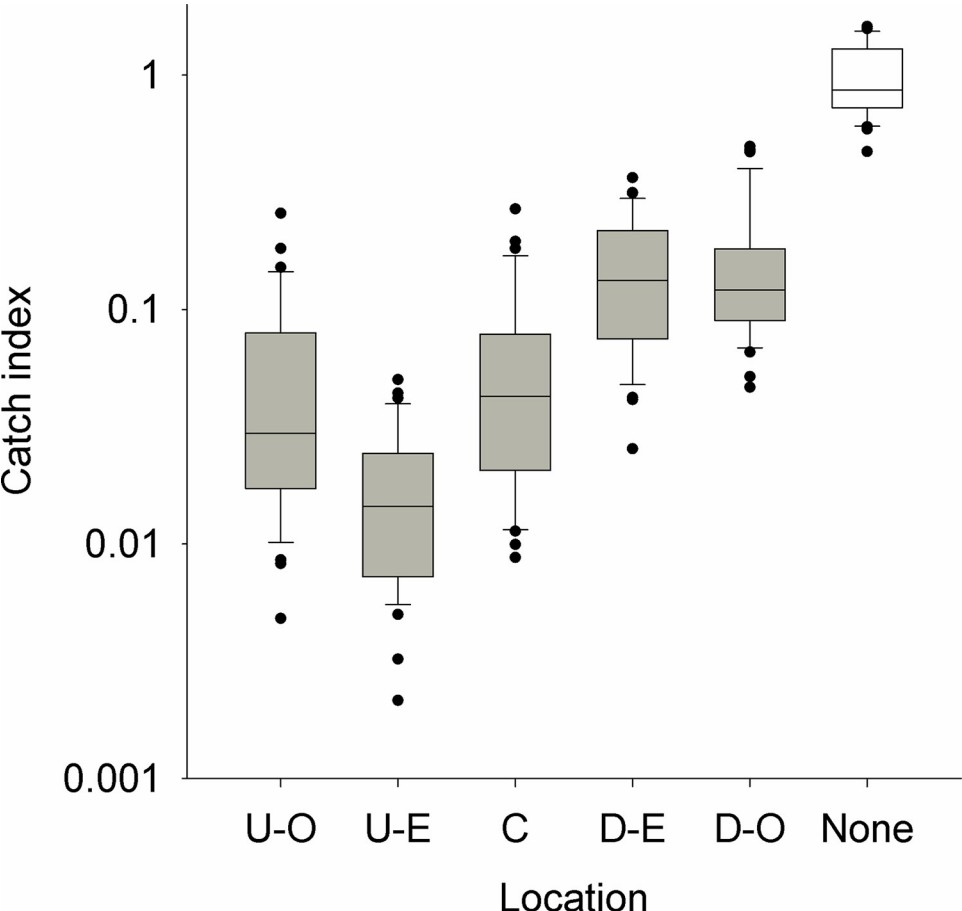

**Fig 3.** Boxplot of catch indices for monitoring traps located across an intervention area where targets were present (grey boxes) or absent (open box). Traps deployed in an intervention areas were located upstream and outside (U-O), on the upstream edge (U-E), centre (C), or downstream edge (D-E) or downstream and outside (D-O). Catch Indices are the mean daily catch per month at each location (Fig 2) for the period January 2012-October 2014, when targets were deployed, expressed as a proportion of the mean catch for September-October 2011 before targets were deployed.

contrast, the median index for traps on the Kochi river where no targets were deployed was 0.86.

The reduction in catches in 2012 after targets were first deployed were 91% and 60% for the traps outside-upstream and outside-downstream respectively (S1A Table), compared to 92% at the centre. These general patterns persisted as Phase 2 was initiated and targets were deployed more widely (2013), with 97% control being achieved upstream, 83% downstream and 98% at the centre. The improved level of control in Phase 2 is because some of the monitoring traps that were outside the Phase 1 blocks were inside the Phase 2 intervention area. While catches declined by >80% in all locations where targets were deployed, we continued to catch tsetse, albeit in low numbers, at all sites. In 2014, catches at all locations increased slightly. This may be related to an incomplete deployment of targets in mid-2014 (S1 Text) but catches of tsetse were significantly less (P<0.001) than the pre-intervention levels.

## Impact of Tiny Targets on neighbouring catchments

Following the deployment of targets on the Ayi, mean daily catches of tsetse declined significantly (S1B Table) from 13.0 tsetse/trap over the period September-October 2011 (months

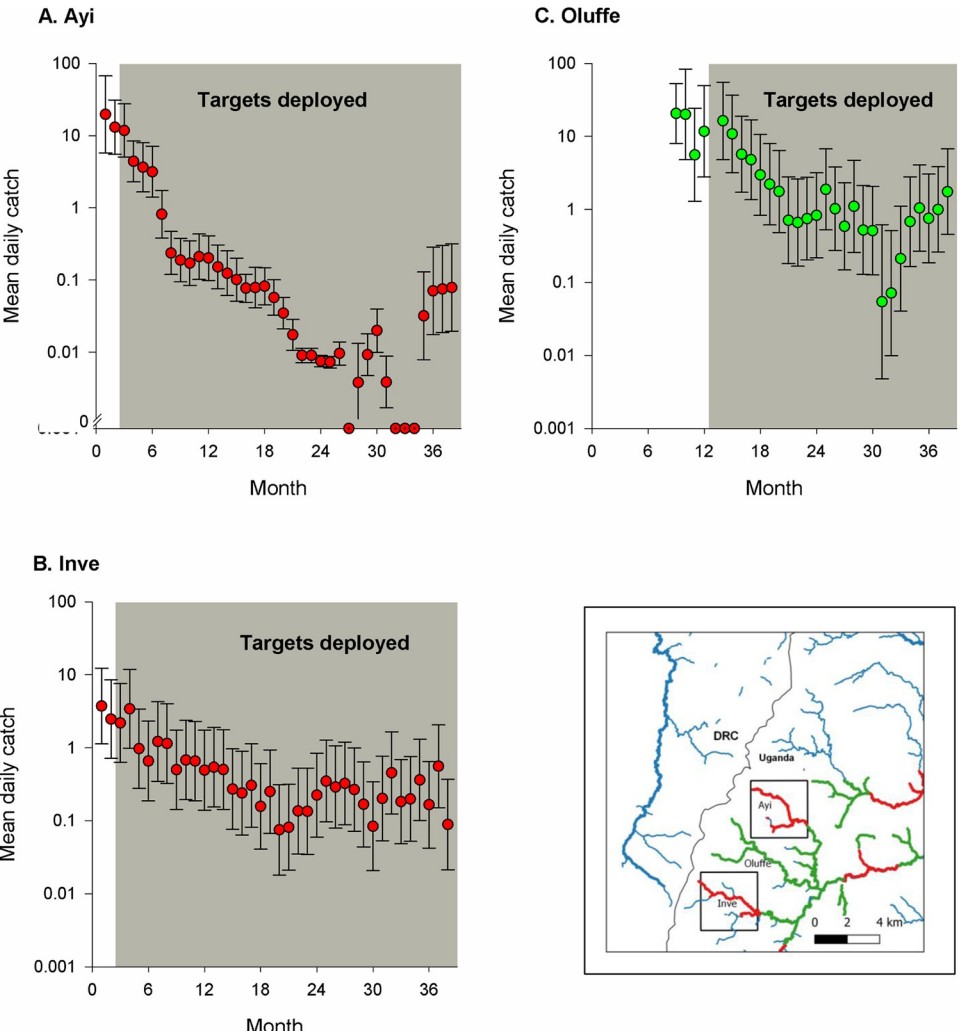

**Fig 4.** Catches of tsetse on the (A) Ayi, (B) Inve and (C) Oluffe rivers. Grey areas denote when targets were present. Inset shows a map of the Ayi, Inve and Oluffe rivers and the distribution of Tiny Targets deployed during Phase 1 (red sections) and Phase 2 (red and green) of the intervention.

1–2) to 0.3 tsetse/trap in 2012 and 0.02 tsetse/trap in 2013–2014 (Fig 4A). Similarly, on the Inve (Fig 4B) mean daily catches declined significantly (S1B Table) from 2.6 in 2012 and 0.2 tsetse/trap in 2013–2014. While mean daily catches averaged 0.3–0.6 in 2012 onwards, those on the adjacent and untreated Oluffe were 40 times greater, averaging 15.9 tsetse/day (Fig 4C) but following the deployment of targets declined significantly (S2B Table) to 1.3 tsetse/day in 2013 and 0.5 tsetse/day in 2014.

## Impact of large-scale tsetse control operations

### Abundance of tsetse

From January 2015 onwards, the area over which Tiny Targets were deployed expanded from ~500 km$^2$ (Fig 1B and 1E) to ~1600 km$^2$ (Fig 1C and 1F). Logistical constraints meant that we were unable to operate traps at a standard frequency in each district every month. Moreover, national holidays associated with Christmas and New Year reduced the number of sampling

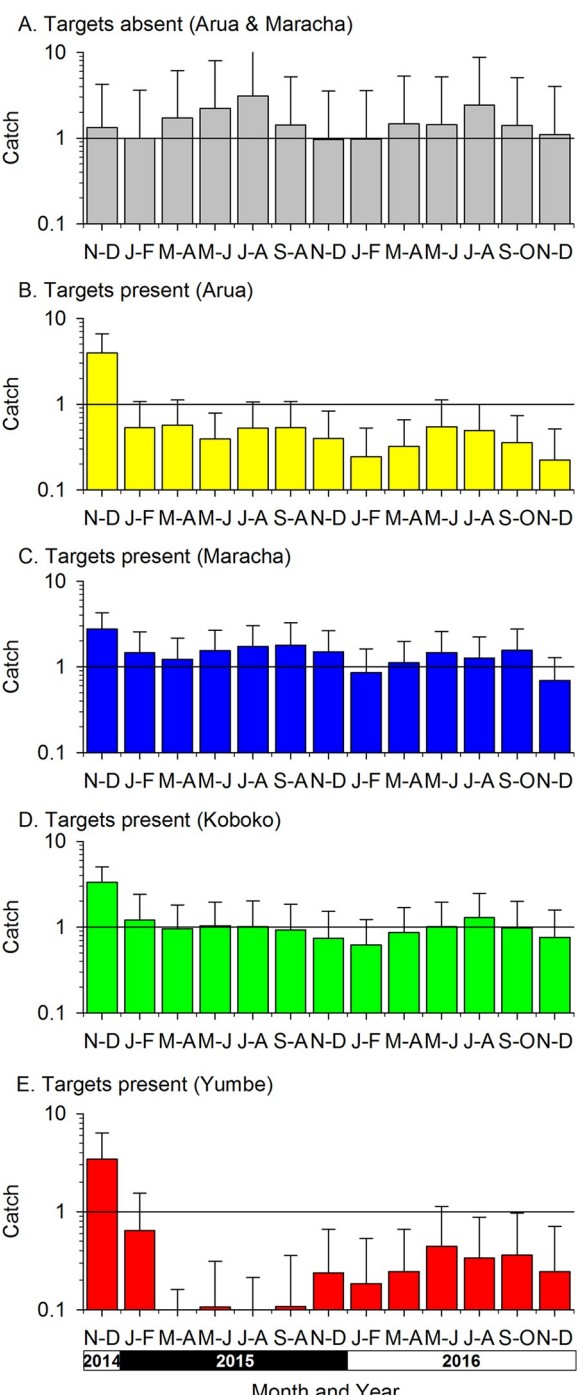

**Fig 5. Mean daily catches of tsetse from monitoring traps in areas with or without Tiny Targets from November 2014 –December 2016.**

days available in December and January. We therefore used two-month periods (i.e., November & December, January & February, etc.) to estimate temporal changes in the abundance of tsetse following the deployment of Tiny Targets.

The results from monitoring traps (Fig 5) show that on rivers in Arua and Maracha districts, at sites 4–14 km upstream of rivers where Tiny Targets were deployed (November

2014-December 2016), the mean daily catch of tsetse remained between1-2 tsetse/trap throughout the monitoring periods (Fig 5A) and there was no significant difference in catches between years (S1C Table). On rivers to which the target deployments were extended, mean daily catches declined significantly in all districts (S1C Table). The decline was most marked in Arua and Yumbe where mean daily catches declined from 3.7 and 3.4 tsetse/trap in 2014 to 0.4 and 0.3 respectively in 2016 (S1C Table). The decline was least in Maracha where the catches declined from 2.8 tsetse/trap in 2014 to 1.2 tsetse/trap in 2016. In comparison to results from 2014, catches in 2016 had declined by 88% in Yumbe and Arua, 66% in Koboko and 52% in Maracha but increased by 26% in areas where targets were absent (S1C Table). Although catches declined in all districts where targets were deployed, nowhere did the catches become zero for extended periods. For instance, four traps located in the original Kubala block of Phase 1, which itself was near the centre of the Phase 3 intervention area (see Fig 1A and 1C), caught a total of 78 tsetse in 136 trap-days in 2016 (mean 0.57 flies per trap day).

**Age structure.** Comparing the ovarian age categories for tsetse collected from rivers where targets were absent (Arua district) or present (Koboko and Maracha) showed that there was no significant difference in age structures in 2015 ($\chi2 = 10.6$, df = 7, P = 0.159) or 2016 ($\chi2 = 11.7$, df = 7, P = 0.112). Analyses of age structure for different districts (Fig 6) showed that for Arua, where targets were not deployed, the age distribution differed significantly between 2015 and 2016 ($\chi2 = 46.1$, df = 7, P<0.001). In 2016, the age distribution in Arua differed significantly from that in Maracha ($\chi2 = 16.3$, df = 7, P = 0.023) and Koboko ($\chi2 = 19.7$, df = 7, P = 0.006). In 2016, the distribution in Arua was also significantly different ($\chi2 = 15.7$, df = 7, P = 0.028) from that in Koboko but not Maracha ($\chi2 = 12.1$, df = 7, P = 0.098).

While there were statistically significant differences in the age structures of populations in areas with or without targets, the numerical differences were small and inconsistent. For instance, the mean ovarian age category of tsetse in Arua (targets absent) and Koboko (targets present) in 2016 was 3.5 and 3.6 respectively. In 2015, the mean ovarian age categories in Maracha and Koboko were greater than that in Arua but the difference was reversed in 2016. Overall, the results do not provide strong evidence that Tiny Targets affected age structure.

**Infection rates.** Analyses of tsetse for the presence of trypanosomes (Table 1) showed that only one tsetse had a salivary gland infection, indicating that the percentage of tsetse with mature *T. brucei* infections was low in all areas. There was no significant effect of year (Deviance = 0.013, df = 1) or district (Deviance = 1.883, df = 2) on the proportion of tsetse with midgut infections. For mouthpart infections however, there was a significant effect of district (Deviance = 15.11, df = 2, P<0.001) and year (Deviance = 5.634, df = 1, P = 0.02) and an interaction between the two (Deviance = 16.44, df = 2, P<0.001). The proportion of tsetse with mouthpart infections in 2015 increased in the order Koboko-Maracha-Arua consistent with Tiny Targets reducing the infection rate. However, in 2016 the order was Arua-Koboko-Maracha in 2016. These analyses suggest that there was not a clear and consistent effect of Tiny Targets on the proportion of tsetse infected with trypanosomes.

**Performance of Tiny Targets.** We inspected the condition of 780 targets across four districts and found that the proportion remaining functional declined with increasing days from deployment (F = 8.5, df = 1, P<0.001); 61% were functioning at four weeks after deployment declining to <20% by 22 weeks (Fig 7A); the median effective life of a Tiny Target was 61 (41.8–80.2, 95% CI) days. There was also significant variation between districts (F = 6.5, df = 3, P<0.001,) with Maracha and Arua having a significantly lower proportion of functioning targets than either Koboko or Yumbe (Fig 7B). There was no significant interaction between the effects of target age and district (F = 1.6, df = 3, n.s.). Overall, 42% (322/775) of targets inspected were functioning, 11% (85) were damaged, 21% (159) had fallen over and 27% (209) were missing.

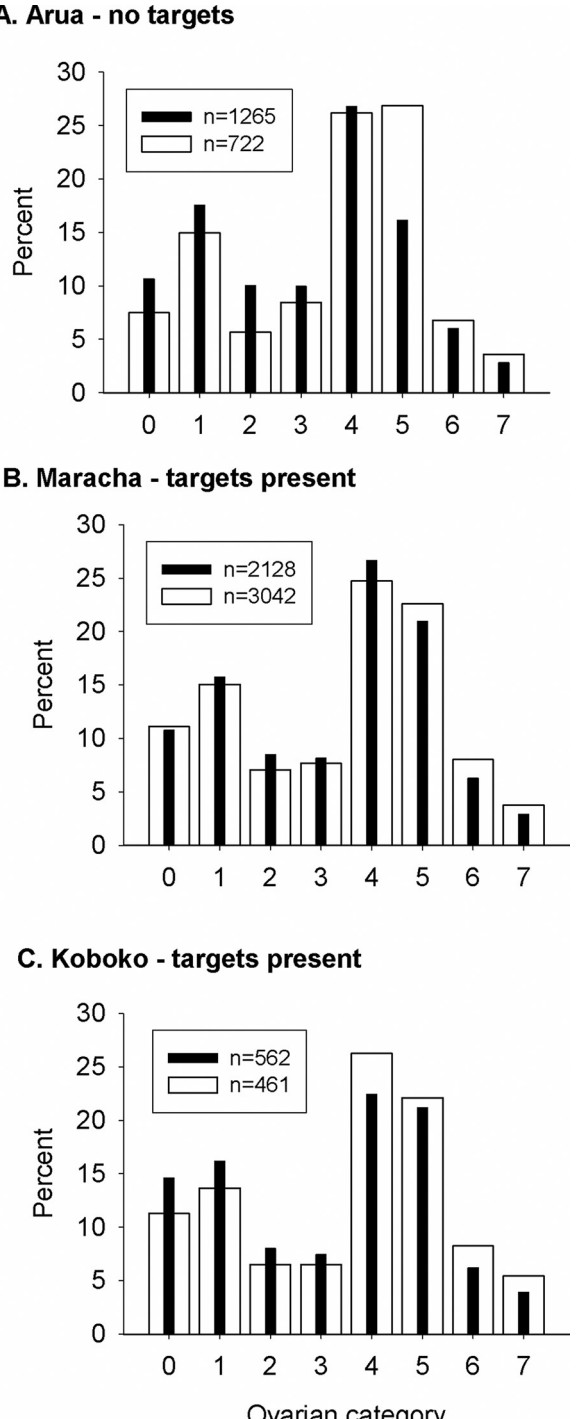

**Fig 6.** Ovarian age structure of female tsetse from sites in (A) Arua district where targets were absent, or (B) Maracha and (C) Koboko districts where they were present. Solid and open bars show data for 2015 and 2016 respectively.

**Table 1. Percentage of tsetse captured in 2015–2016 with trypanosomes observed in salivary glands, midgut or mouthparts from areas with targets present (Koboko, Maracha) or absent (Arua).**

| Year | District | n | Salivary glands | | Midgut | | Mouthparts | |
|------|----------|---|------|--------|------|--------|------|--------|
| | | | Mean | 95% CI | Mean | 95% CI | Mean | 95% CI |
| 2015 | Arua | 1118 | 0.00 | 0.000–0.329 | 0.27 | 0.055–0.782 | 1.25 | 0.686–2.092 |
| | Koboko | 502 | 0.00 | 0.000–0.732 | 0.20 | 0.005–1.105 | 0.00 | 0.000–0.732 |
| | Maracha | 1944 | 0.00 | 0.000–0.190 | 0.21 | 0.056–0.526 | 1.08 | 0.670–1.647 |
| 2016 | Arua | 663 | 0.00 | 0.000–0.555 | 0.00 | 0.000–0.555 | 0.15 | 0.004–0.837 |
| | Koboko | 421 | 0.00 | 0.000–0.872 | 0.71 | 0.147–2.068 | 0.71 | 0.147–2.068 |
| | Maracha | 2761 | 0.04 | 0.001–0.202 | 0.18 | 0.059–0.422 | 2.10 | 1.599–2.707 |

## Discussion

### Impact within drainage systems

Monitoring tsetse across a small-scale intervention showed, not surprisingly, that within a drainage system the number of tsetse declined by >90% in the section where targets were deployed. More surprisingly, good control was also obtained on the edge and outside of the intervention section, especially in the upstream parts of the system (Fig 3). For instance, after only one year of control, number of tsetse had declined by 91% upstream of the intervention and 60% downstream (S1A Table). These patterns are probably related to the natural movement and abundance of tsetse. Analyses of factors affecting the abundance of riverine tsetse have shown that in this part of Uganda catches of tsetse are strongly correlated with the overall density of rivers and associated vegetation [22]. In upper parts of the river network, streams are smaller and seasonal, and associated vegetation is reduced. Consequently, tsetse abundance is lower and fewer tsetse can re-invade an intervention area.

### Impact across drainage systems

While we found that catches declined in contiguous parts of the same river where targets were not deployed (Figs 2 and 3), our results suggest that Tiny Targets did not control tsetse on

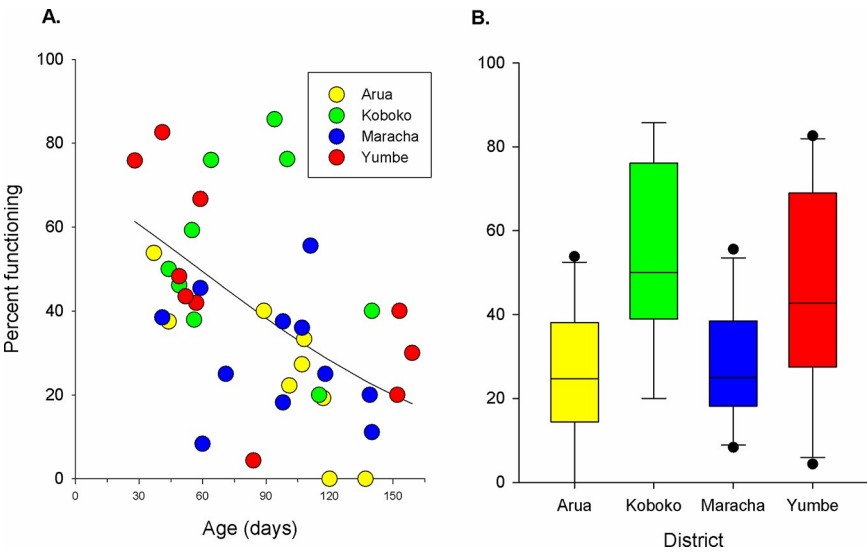

**Fig 7.** (A) Scatterplot of percentage of targets functioning against age (days since deployment) and (B) boxplot of percentage functioning for each district.

adjacent rivers. The deployment of targets on the Ayi and Inve rivers seemed not to impact on the Oluffe river which lies between them (Fig 4, inset). However, catches on the Oluffe declined by >90% following the deployment of targets (Fig 4C). Previous studies with riverine [22] and savanna [40] tsetse have indicated that targets reduce the abundance of the flies outside the intervention area. The finding that the impact of Tiny Targets is mostly along rivers rather than between them accords with studies of marked tsetse which showed that few riverine tsetse moved between rivers [23].

The evidence that deployment of targets on the Ayi and Inve rivers did not impact on tsetse populations on the Oluffe assumes that the densities on these three tributaries of the Enyau were broadly similar before targets were deployed. Evidence to support this assumption is provided by the results of a tsetse survey conducted in this area in October-November 2010, before targets were deployed, which showed that catches from traps deployed on these rivers were similar [22]. This finding suggests that tsetse populations may persist if targets are not deployed on all rivers.

The present results also provide first information on the impact of removing targets on a tsetse population. The traps in areas without targets in Phase 3 were located in parts of Arua and Maracha where targets had been present in Phases 1–2. The mean daily catches in these areas, after cessation of control, ranged between 1.0 and 3.1 tsetse/trap (Fig 5A). The traps in Maracha were deployed on Oluffe and caught 5.6–20.6 tsetse/trap before targets were deployed (Fig 4C) and those in Arua caught 9.0 tsetse/trap [13]. These results suggest that withdrawal of targets from western/upstream parts of Arua and Maracha in January 2015 did not result in tsetse recovering rapidly to their pre-intervention levels during the 24 month sampling period (January 2015 –December 2016).

Models of tsetse populations re-bounding following complete cessation of control suggest that the population will recover to ~50% of its pre-intervention abundance after 24 months [13]. We suggest that the tsetse population did not recover as rapidly as models predict because the large Phase 3 intervention in the downstream areas reduced the numbers of tsetse, preventing re-population of the upstream sections.

## Impact of large-scale control on abundance, age structure and infection rates

The development of target deployment from a small-scale trial, managed and implemented by researchers, to a large-scale operation managed by a national programme and implemented by District Entomologists was successful; targets were deployed regularly at six month intervals covering most of the intended 1600 km$^2$.

In accord with previous studies of the impact of Tiny Targets on *G. f. fuscipes* in Uganda [13], we show that the deployment of targets in new districts (Yumbe, Arua) led to a >80% decline in the catches of tsetse from monitoring traps. Similarly, catches from monitoring traps deployed in Koboko also declined by >60% but the decline was less marked in Maracha (52%). Prior to the deployment of targets, the mean daily catch from traps deployed in Maracha was 2.8 tsetse/day and catches in the following months ranged between 0.7 and 1.8 tsetse/day, representing declines of between 75% and 46% respectively (Fig 5). The smaller effect in Maracha may be due, in part, to the impact of earlier deployments of Tiny Targets during Phases 1 and 2, i.e., operations conducted in 2011–2014. These covered large parts of Maracha district (see Fig 1) and previous analyses of this intervention [13] showed, for instance, that the mean daily catches from monitoring traps operated in the Kubala block of Maracha district (Fig 1A) caught 12.4 tsetse/trap in September 2011, before targets were deployed. Catches subsequently declined to 0.3 tsetse/trap by June 2012 [13]. In the present study the catches from

traps within the original Kubala block caught a total of 78 tsetse over 136 trap-days (arithmetic mean = 0.6 tsetse/trap/day) in 2016. Hence comparing the catches in 2015–2016, when targets were present, against catches in late 2011, suggests that there has been a >90% decline in the abundance of tsetse since Tiny Targets were first deployed.

While the catches of tsetse from monitoring traps declined following the deployment of targets, the age structure of the tsetse population did not change. This contrasts with findings in Zimbabwe where deployment of odour-baited targets had a marked effect on the age structure of a tsetse population [25]. Six months after the deployment of targets, the mean age class was 0.8–1.8 compared to 3.8 before targets were deployed. Analyses of the data from Zimbabwe suggested that mean daily mortality was between 4–10% where targets were present compared to 1–3% in their absence. Our estimates are based on dissections of 8308 tsetse compared to 1069 in the Zimbabwe-based study and hence it seems unlikely that the absence of any effect in the present study is due to a small sample size. A limitation in our study of age structure is that tsetse were not collected from the centre of the intervention area where we would expect the highest level of control and low levels of invasion. However, the low numbers of tsetse caught from these areas (e.g., 78 tsetse caught in the Kubala block in 2016) hampers robust estimates of age structure.

For all phases of the intervention, the catch of tsetse from monitoring traps generally declined to ≤1 tsetse/trap/day but did not reach and remain at zero in any parts of the intervention area. This can be explained most plausibly by invasion of tsetse from neighbouring areas where targets were not deployed. The fact that Tiny Targets caused a sharp decline in the density of tsetse populations but did not eliminate them completely, or change their age structure and infection status, confirmed the expectation that invasion can occur from either tsetse-infested areas outside or untreated drainages within the intervention area. The capture of large numbers of old flies (ovarian category >4) is consistent with invasion, since it takes time for tsetse to invade.

## Effective life of Tiny Targets

The persistence of low numbers of tsetse across the intervention area may have been assisted by the effective life of the targets being lower than predicted. The effective life of Tiny Targets was reported to be six months in a small-scale (<1 km2) field trial on an island in Lake Victoria [13]. In the present study, the survival of targets in the field was much less. In Maracha for instance, where catches persisted at about ~1 tsetse/trap/day, only 25% of targets inspected were functional (Fig 7). Previous modelling of the impact of Tiny Targets [13] suggested that the mortality imposed by targets was ~4%/day and hence an 80% reduction in the density of functioning targets would reduce this to ~1%/day which is below the imposed mortality (4%/day) required to eliminate tsetse [12]. Nonetheless, imposing a mortality of 1%/day is predicted to reduce tsetse populations by about 90% but not lead to local elimination of tsetse [41]. However, the elimination of tsetse is not a pre-requisite for the elimination of gHAT. Mathematical models of gHAT [3,4,13,42] and empirical studies [4, 16,20] suggest that >70% reduction in abundance of tsetse will interrupt transmission.

Rivers in North West Uganda are prone to seasonal floods following heavy rain and these may be an important cause of the 27% of targets missing. We have also noticed that during the dry season, livestock visiting the river to drink also knock over targets. Some of these losses might be mitigated by placing the targets away from livestock watering points or on elevated places less vulnerable to flooding. Measures to prolong the effective life of targets must however be balanced by the need to ensure that they are in places where tsetse are present. The persistence of low numbers of tsetse across the Phase 3 intervention area may also be a result of the patchy distribution of targets. As is clear in Fig 1F, the targets were deployed along only the

larger rivers (>2 m wide) and it may be that small self-sustaining populations of tsetse persist within the intervention area.

## Limitations

The present analysis aimed to use monitoring data from a national control programme to answer questions about the impact of scale-up on tsetse control. The scaling up and monitoring were not designed explicitly to address these research questions. Stronger inferences and conclusions could be produced by designing trials and sampling frameworks for this specific purpose [43]. For instance, the evidence that interventions in one tributary do not affect tsetse in an adjacent one could be strengthened by repeating the study across several paired rivers and designing the sampling framework so that each river was sampled simultaneously before and after targets were deployed, and by considering additional factors such as the distance between perennial rivers. Conclusions about the impact of target operational longevity on levels of control could be tested by comparing different densities of target with frequent monitoring of the targets to ensure that the effective densities are quantified. Monitoring of tsetse populations was carried out on the rivers where targets were deployed and we did not quantify the impact on tributaries and smaller streams. Tsetse populations on these smaller rivers are likely to be less affected by the deployment of Tiny Targets and hence the overall reduction in the tsetse population may be less than that inferred by catches from our monitoring traps.

## Conclusions

In respect of the four questions posed in the Introduction, our studies offer the following answers.

1. Current practices for deploying Tiny Targets in Uganda reduce the density of tsetse by >80%, which can contribute to a sustained reduction in the incidence of gHAT [20].

2. Control of tsetse in one part of a river line can reduce the tsetse population for several kilometres on the adjacent lengths of the same river, but it has no detectable effect on tsetse in other rivers a kilometre or so away.

3. Expansion of the operational area did not always produce higher levels of suppression and did not produce any detectable change in the age structure of the population, perhaps due to the failure to treat the smaller streams and/or invasion from adjacent untreated areas.

4. The lifespan of targets was very variable but was typically no longer than a few months.

   The implication of these answers is that even better control of tsetse might be achieved by (i) deploying targets in places where they are less likely to be damaged by livestock and seasonal floods and (ii) deploying targets on more rivers, including small streams. However, the benefits of better control need to be balanced against the increased costs of deploying more targets. In the context of Uganda, where incidence of gHAT is exceedingly low, deploying targets on smaller rivers may not be cost-effective.

## Supporting information

**S1 Text. Supplementary Materials—1: Biannual vs. Annual Deployment of Tiny Targets.** S1 Text: Supplementary Materials—Fig S1.1 Mean daily catch of tsetse in January-October 2014 in areas with (A) no targets, (B) annual deployment or (C) biannual deployment of targets. S1 Text: Supplementary Materials—Fig A. Schematic map of the arrangement of monitoring traps (red triangles) deployed in a transect along a drainage system (blue line) to

quantify the impact of small-scale deployment of Tiny Targets. Dotted green line indicates the limits of a 7 x 7 km block within which targets were deployed in Phase 1. S1 Text: Supplementary Materials—Table A. Mean daily catch (95%CI) of tsetse from monitoring traps deployed along rivers transecting where Tiny Targets were deployed. S1 Text: Supplementary Materials—Table B. Mean daily catch (95%CI) of tsetse from monitoring traps deployed on Inve, Ayi and Oluffe rivers. S1 Text: Supplementary Materials—Table C. Mean daily catch (95%CI) of tsetse from monitoring traps deployed in areas with or without Tiny Targets.
(DOCX)

## Acknowledgments

We thank the District Entomologists and Animal Production Officers, particularly Tom Onzivua and Stephen Onzima (Koboko), Harriet Batreru (Maracha), Philliam Cema (Arua) and Rashid Kawawa (Yumbe) and their teams who deployed Tiny Targets. We also thank the field technicians, Henry Ombanya, Mathia Amayo Erindia, Ignatius Jurua, Mark Ojoma and Patrick Edema who operated the network of monitoring traps, and the laboratory technicians, Victor Drapari, Edward Aziku and Alex Trima who performed the ovarian and infection dissections.

## Author Contributions

**Conceptualization:** Andrew Hope, Albert Mugenyi, Johan Esterhuizen, Inaki Tirados, Lucas Cunningham, Mike J. Lehane, Steve J. Torr, Glyn A. Vale, Charles Waiswa, Richard Selby.

**Data curation:** Andrew Hope, Albert Mugenyi, Johan Esterhuizen, Inaki Tirados, Lucas Cunningham, TN Clement Mangwiro, Michelle Stanton, Steve J. Torr, Richard Selby.

**Formal analysis:** Andrew Hope, Gala Garrod, Joshua Longbottom, Michelle Stanton, Steve J. Torr, Glyn A. Vale.

**Funding acquisition:** Mike J. Lehane, Steve J. Torr.

**Investigation:** Andrew Hope, Albert Mugenyi, Johan Esterhuizen, Inaki Tirados, Lucas Cunningham, TN Clement Mangwiro, Mercy Opiyo, Steve J. Torr, Richard Selby.

**Methodology:** Andrew Hope, Lucas Cunningham, Michelle Stanton, Steve J. Torr.

**Project administration:** Andrew Hope, Albert Mugenyi, Johan Esterhuizen, Inaki Tirados, Mike J. Lehane, TN Clement Mangwiro, Steve J. Torr, Charles Waiswa, Richard Selby.

**Supervision:** Andrew Hope, Albert Mugenyi, Johan Esterhuizen, Inaki Tirados, Lucas Cunningham, Mike J. Lehane, TN Clement Mangwiro, Mercy Opiyo, Steve J. Torr, Richard Selby.

**Validation:** Andrew Hope.

**Visualization:** Michelle Stanton, Steve J. Torr, Glyn A. Vale.

**Writing – original draft:** Andrew Hope, Steve J. Torr, Glyn A. Vale.

**Writing – review & editing:** Albert Mugenyi, Johan Esterhuizen, Inaki Tirados, Lucas Cunningham, Gala Garrod, Mike J. Lehane, Joshua Longbottom, TN Clement Mangwiro, Mercy Opiyo, Michelle Stanton, Charles Waiswa, Richard Selby.

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
