## [Decision Letter · Decision Letter 0]

14 Mar 2022

Dear Dr. Torr,

Thank you very much for submitting your manuscript "Scaling up of tsetse control to eliminate Gambian sleeping sickness in northern Uganda" for consideration at PLOS Neglected Tropical Diseases. As with all papers reviewed by the journal, your manuscript was reviewed by members of the editorial board and by several independent reviewers. The reviewers appreciated the attention to an important topic. Based on the reviews, we are likely to accept this manuscript for publication, providing that you modify the manuscript according to the review recommendations. 

Sincerely,

Adly M.M. Abd-Alla, Prof asso.

Associate Editor

Jan Van Den Abbeele

Deputy Editor

Reviewer's Responses to Questions

**Key Review Criteria Required for Acceptance?**

**Methods**

-Are the objectives of the study clearly articulated with a clear testable hypothesis stated?

-Is the study design appropriate to address the stated objectives?

-Is the population clearly described and appropriate for the hypothesis being tested?

-Is the sample size sufficient to ensure adequate power to address the hypothesis being tested?

-Were correct statistical analysis used to support conclusions?

-Are there concerns about ethical or regulatory requirements being met?

Reviewer #1: -Are the objectives of the study clearly articulated with a clear testable hypothesis stated? Yes

-Is the study design appropriate to address the stated objectives? Yes

-Is the population clearly described and appropriate for the hypothesis being tested? Yes

-Is the sample size sufficient to ensure adequate power to address the hypothesis being tested? Yes

-Were correct statistical analysis used to support conclusions? Generally, but see specific small comments.

-Are there concerns about ethical or regulatory requirements being met? No

Reviewer #2: The objectives of the study is clearly articulated and described.

Reviewer #3: There are some editorial issues, these are detailed below.

**Results**

-Does the analysis presented match the analysis plan?

-Are the results clearly and completely presented?

-Are the figures (Tables, Images) of sufficient quality for clarity?

Reviewer #1: -Does the analysis presented match the analysis plan? Yes

-Are the results clearly and completely presented? Yes

-Are the figures (Tables, Images) of sufficient quality for clarity? Yes

Reviewer #2: It is stated in the under the “Entomological monitoring” section that “For Phases 1 and 2, 14 traps were also deployed along the Kochi river in Koboko district (northern most traps in Fig 1, A & B) where no targets were deployed. These traps provided a measure of the underlying seasonal variation in the abundance of tsetse in the absence of Tiny Targets.” When the results of Phases 1 and 2 are given no mention is made regarding the abundance of flies form this site. The site is only discussed in relation to the Phase 3. Some information of the appearance of tsetse before intervention during the time of Phases 1 and 2 should be discussed. 

In the section “Statistical analyses” it is stated that statistical models were used to assess catch data and proportion of target functioning correctly. Statistical analyses are only given for the condition of the Tiny Targets and not for the catch data. The catch data analyses need to be given and discussed.

Reviewer #3: These are likely completed correctly, but the writing needs improvement in organization to make this adequate for publication. See details below.

**Conclusions**

-Are the conclusions supported by the data presented?

-Are the limitations of analysis clearly described?

-Do the authors discuss how these data can be helpful to advance our understanding of the topic under study?

-Is public health relevance addressed?

Reviewer #1: -Are the conclusions supported by the data presented? Yes

-Are the limitations of analysis clearly described? Yes

-Do the authors discuss how these data can be helpful to advance our understanding of the topic under study? Yes

-Is public health relevance addressed? Yes

Reviewer #2: The findings are valuable as they improvement on our knowledge on innovative tools to be used in tsetse control programs. The results are clearly presented and well discussed.

Reviewer #3: The conclusion paragraph is not appropriate, it is not limited to findings from this study and does not address the major questions set out to answer.

**Editorial and Data Presentation Modifications?**

Reviewer #1: Yes, and I have listed these above, and/or noted them on the attached pdf

Reviewer #2: The manuscript is well written

Reviewer #3: I outline the changes I think necessary below.

**Summary and General Comments**

Reviewer #1: This is a fairly straightforward paper that reports the continuing, and successful, use of “Tiny Targets” to produce levels of control of tsetse (Glossina sp) in Uganda, in efforts to control and eventually eliminate Gambian sleeping sickness. As such the paper is an important contribution, detailing the successful use of vector control in the fight against Gambian HAT, and should be published.

I have no major criticisms – and my minor comments are all added as edits on the pdf (find attached).

Just be clear about a couple of small niggles that do need to be addressed:

1. The authors need to think more carefully about why it is possible to get away with "sparse deployment of baits". To my mind this is not simply because of the dispersal propensity of tsetse. It is also a function of the extremely low rate of reproduction in tsetse - which means that we only need to kill 3.5% of adult females per day in order to send the population into negative growth. Rewrite the appropriate text, referring to the appropriate papers. 

2. The authors seem to get confused between “square km” [km2] and “per square km” [km-2].

Please check throughout.

3. Please do a proper statistical analysis of the data in Table 1, providing estimates, 95% confidence intervals and appropriate statistical analyses, and an interpretation of is happening.

4. On the statistics front, it is also insufficient to provide an F statistic and a P value: the relation between the two is a function of the degrees of freedom, which must therefore be stated.

The paper is generally well written and presented. 

Also attached is a copy of the pdf of the paper – with edits attached that the authors should address.

Reviewer #2: The manuscript presents data on the use of insecticide impregnated “Tiny Targets” to control Glossina fuscipes fuscipes in Uganda. The manuscript focuses on the impact of small-scale interventions as well as the impact of wen these small-scale interventions are scaled up. Additionally, the live time of the Tiny Targets is assessing withing the Uganda experience. Publications of operational data is very important for future dissection making for vector control. This manuscript will add to the knowledge in this field.

Reviewer #3: Hope and colleagues use trapping data from monitoring traps within and around five Tiny Target intervention areas in Uganda 2011-2016 to test for an effect of increasing intervention area from ~250 km2 to ~1600 km2. Specifically, the study sets out to test: (1) the impact of small-scale interventions on neighboring untreated areas, (2) if expansion produced higher levels of suppression within the control area, (3) the impact of intervention on age structure and infection rates (4) the impact of treated tributaries on a neighboring untreated tributary, and (5) the operational lifetime of Tiny Targets on this larger scale program. Their results indicated (1) an 80% decline in tsetse abundance, with greater reduction in abundance upstream rather than downstream interventions (but what scale was this at? Small or large? I think both… but it is a little unclear how the result relates back to the stated questions/goals), (2) successful expansion, (3) no impact on age structure or infection rates, (4) no spill-over effect on the neighboring tributary, and a shorter-than-expected operational lifetime of the Tiny targets.

Major comments:

1) The intro does not introduce all concepts necessary to understand the importance of some components of the study. Several of the following points are examples of this. The importance and implications of up vs downstream measurements is unclear, more explanation is needed in the intro and discussion to make it clear why you measure this, and what the results mean.

2) Especially in the intro, statements about 70% suppression being sufficient to break the transmission cycle are not well explained or supported. Even after reading the discussion where 6 papers are cited, I am not convinced that this threshold is broadly well-supported. This is a complicated issue, with some not-so-intuitive responses in infection rate. For instance, isn’t there a reduction in dilution effect with fewer flies, which would lead to short-term increased infection rates very locally around the infected hosts? Furthermore, results of infection rates do not support this assertion, yet it comes up again in the conclusion repeating the statement as if it were a fact. I think this issue needs to be much more carefully stated from the beginning, with the source of the statement clearly laid out with caveats acknowledged (even in the abstract). Then it should be removed from the conclusion, and you should replace this with conclusions drawn from the present study.

3) My sense is that this 70% suppression threshold is from mathematical models with some empirical evidence in some settings, but real-world stochasticity and spatial heterogeneity alters this threshold and the dynamics of this variation have not yet been adequately studied. If this is true, then the logic should be put to use to explain the importance of testing this threshold, and your data can inform this. In my opinion, predictions of infection rates should be more clearly motivated in the intro, and testing the prediction of reduced infection rates should be included in the stated goals (otherwise, why do you go through the trouble of estimating them?). 

4) On this topic, I suspect that there is a lot of interesting detail in your results that was lost when you averaged across the full study area for Table 1. Can you separate by drainage? Can you even compare infection rates in different areas of suppression (inside+edge vs outside for example?). This would be really interesting! And why did you split everything up by body parts? There is probably really interesting reasons for doing this, so can you explain and include these details? Does separating by drainage help explain the mismatch of infection status in midgut vs mouthparts? Or are their other caveats such as miss-scoring of any of these body parts that might explain the mismatch? I would find these results and discussions very interesting. I know that you are dealing with small sample sizes, but with the data I can see (original data does not specify which drainage or which treatment area the flies are from for Table 1), it seems like there could be more meaningful interpretation can be made.

5) In the intro, there is no clear set-up of why age structure is expected to change (this is buried in the discussion and should be moved to the intro). Further, the prediction is lumped in with the stated goal of how the expansion of tsetse control operations impacted levels of suppression. You would be better off dedicating a paragraph in the intro and a stand-alone research question. 

6) The discussion does not clearly answer the questions laid out in the intro and methods. This problem is especially evident in the conclusion, which does not address the central issue presented in the summary, abstract and intro (expansion of the intervention area). I would encourage the authors to build a coherent outline that is then addressed in each section: intro, methods, results, discussion. Usually this works best if you keep the order logical and consistent (although exceptions can flow better, depending on the details, especially in the discussion).

7) The methods are not complete. Where is the description of how a “functional trap” is defined and measured? Where is the info about how age was determined, and details on how infection status was determined in each body part? Most importantly, the different statistical tests performed to answer each question are not clear. I would strongly advise stating the methods and predictions for each question/hypothesis, including the two that are missing sub-headings (age structure and infection status). Then the section describing the statistical methods can be a short section describing what was done when not otherwise noted and graphical methods, or it can be deleted. Finally, when reporting the “significance”, the statistical test performed, relevant settings and sampling design, and relevant results (such as p-values, etc.) should always be reported along with the figure and table with the data that was tested. Even in the discussion, the data used and result referenced should be apparent when you discuss each result. Add references to the appropriate figures and tables throughout.

8) There is a big issue here in study design because there is no control for any of the tests. Each of the comparisons made is confounded by differences in time or space. If I am understanding your experimental set-up correctly, there are no “centers” of treatment that did not get expanded treatment, so there is no control for the main question of the study. Thus, the centers have differences in time since beginning treatment as well as being in the center There is a similar problem (this time confounded by time) in comparison of the drainage with/without treatment, the no-treatment drainage was not sampled pre-treatment along with the other drainages, so it cannot serve as a true control. I see that you put a section that discusses limitations, and although I think this can be a part of your strategy to deal with these issues, I do not think it is sufficient on its own. I think you need to add details of the limitations for each research question right up-front where you discuss the experimental design and the statistical tests used. This way the reader can understand what is the appropriate interpretation of each of your results. 

Minor comments:

9) Mapping methods for rivers. What are you sources and methods? For example, how do you make the major rivers thicker lines on the maps than the minor tributaries? In my understanding, this is not simple, can you give all details to make it reproducible for future researchers?

10) Line numbers missing. I would have some grammatical and editorial comments if there were line numbers, next time make sure you add those to your document! I am unable to spend the time explaining the location of each of my edits without them at this time.

11) It would be good to see a report of the deployment data, this would allow for discussion of “incomplete deployment of targets in mid-2014”. As written, this comes out of nowhere and is not supported by data.

12) Are 1 and 0.1 catches per day biologically relevant? If so explain, if not, then I would encourage you to use more descriptive language that describes the actual levels rather than saying above/below these arbitrary thresholds. Additionally, are the “general patterns” described statistically significant differences? It is not clear from looking at the figure how significantly different they may or may not be, and then the description using the thresholds makes it possible these are arbitrary differences that are not statistically or biologically significant.

13) I would like to see a paragraph describing general trapping results before addressing the tests to your hypotheses. This would improve the flow and the readability.

14) There are several statements in the results that should be put in the discussion, and are also do not logically flow. 

a. “The higher catches on the Oluffe suggests…” Given limitations, this statement is basically speculation and belongs in the discussion with caveats. Further, it is stated before the relevant results are presented.

b. “Overall, the results suggest that the effects of targets…” Belongs in the discussion.

PLOS authors have the option to publish the peer review history of their article (what does this mean?). If published, this will include your full peer review and any attached files.

Reviewer #1: No

Reviewer #2: No

Reviewer #3: No

Figure Files:

Data Requirements:

Reproducibility:

References

---

## [Editor Report · Decision Letter 1]

23 May 2022

Dear Dr. Torr,

We are pleased to inform you that your manuscript 'Scaling up of tsetse control to eliminate Gambian sleeping sickness in northern Uganda' has been provisionally accepted for publication in PLOS Neglected Tropical Diseases.

Best regards,

Adly M.M. Abd-Alla, Prof asso.

Associate Editor

Jan Van Den Abbeele

Deputy Editor

---

## [Editor Report · Acceptance letter]

23 Jun 2022

Dear Dr. Torr,

We are delighted to inform you that your manuscript, "Scaling up of tsetse control to eliminate Gambian sleeping sickness in northern Uganda," has been formally accepted for publication in PLOS Neglected Tropical Diseases.

Best regards,

Shaden Kamhawi

co-Editor-in-Chief

Paul Brindley

co-Editor-in-Chief
